# Regulation of Age-Related Lipid Metabolism in Ovarian Cancer

**DOI:** 10.3390/ijms26010320

**Published:** 2025-01-01

**Authors:** Jihua Feng, Clay Douglas Rouse, Lila Taylor, Santiago Garcia, Ethan Nguyen, Isabella Coogan, Olivia Byrd, Andrew Berchuck, Susan K. Murphy, Zhiqing Huang

**Affiliations:** 1Division of Reproductive Sciences, Department of Obstetrics and Gynecology, Duke University School of Medicine, 701 West Main Street, Suite 510, Duke, P.O. Box 90534, Durham, NC 27701, USA; jihua.feng@duke.edu (J.F.); lila.taylor@duke.edu (L.T.); santiago.garcia@duke.edu (S.G.); ethan.nguyen@duke.edu (E.N.); isabella.coogan@duke.edu (I.C.); olivia.byrd@duke.edu (O.B.); susan.murphy@duke.edu (S.K.M.); 2Department of Emergency, The Second Affiliated Hospital, Guangxi Medical University, Nanning 530021, China; 3Division of Laboratory Animal Resources (DLAR), Duke University School of Medicine, Durham, NC 27701, USA; clay.rouse@duke.edu; 4Division of Gynecologic Oncology, Department of Obstetrics and Gynecology, Duke University School of Medicine, Durham, NC 27701, USA; andrew.berchuck@duke.edu

**Keywords:** ovarian cancer, aging adipose microenvironment, tumor microenvironment, free fatty acids, omega-5, *S100a8*, s100a9, rat ovarian cancer xenografts

## Abstract

The mortality rate of ovarian cancer (OC) remains the highest among female gynecological malignancies. Advanced age is the highest risk factor for OC development and progression, yet little is known about the role of the aged tumor microenvironment (TME). We conducted RNA sequencing and lipidomic analysis of young and aged gonadal adipose tissue from rat xenografts before and after OC formation. The rates of tumor formation (*p* = 0.047) and tumor volume (*p* = 0.002) were significantly higher in the aged rats than in their young counterparts. RNA sequencing data showed significant differences in gene expression profiles between the groups of young and aged rat adipose tissues (*p* < 0.05), including *S100a8*, *S100a9*, *Il1rl1*, *Lcn2*, *C3*, *Hba-a1*, *Fcna*, and *Pnpla3*. At the time of tumor generation, there were also changes in the lipid components within the gonadal adipose tissues of young and aged rats, with higher levels of free fatty acids (FFAs) and triglycerides (TGs) in aged rats. Furthermore, the aged TME showed changes in immune cell composition, especially inflammation-related cells, including neutrophils, myeloid dendritic cells, CD4+ T cells (non-regulatory), and mast cell activation (*p* < 0.05). The correlation between *S100a8*, *S100a9*, neutrophil, and omega-5, FFA 18:3 levels was also determined. Additionally, omega-5, which is downregulated in aged rats, inhibited OC cell proliferation in vitro (*p* < 0.001). Our study suggests that the aged TME promotes OC proliferation resulting from age-related changes in gene/pathway expression, lipid metabolism, and immune cell distribution. Targeting the aging adipose microenvironment, particularly lipid metabolism, is a promising therapeutic strategy for OC and warrants further investigation. **Significance**: The aging microenvironment contributes to OC development and progression because of changes in the immune response regulatory genes *S100a8* and *S100a9*, secreted by adipocytes, preadipocytes, or neutrophils, and by altering omega-5 metabolism.

## 1. Introduction

Ovarian cancer (OC) is the eighth most common cancer in women globally and has the highest mortality rate among gynecological malignancies, with 324,398 new cases and 206,839 deaths in 2022 [1]. As OC tends to advance rapidly and exhibits no obvious signs or symptoms in its early stages, more than two-thirds of patients are diagnosed at an advanced stage [2]. OC is most frequently diagnosed in women aged 55–64, with a median age of 63 years at diagnosis. The two major factors that affect prognosis and survival are metastasis and recurrence, which are positively related to aging [3,4,5]. Furthermore, older patients have poorer prognoses, as unfavorable progression-free survival and overall survival are associated with older age [6]. Although treatment disparities in the elderly may contribute to this difference, older individuals may also be more susceptible to disease progression due to metabolic and immune changes associated with aging. However, the mechanisms underlying the influence of age on OC development, progression, and recurrence are poorly understood.

OC seems to preferentially spread to adipose tissue within the peritoneal cavity, specifically the omentum, in which the mechanisms related to the proliferation of metastatic OC cells in the intra-abdominal environment are not fully understood. Some studies have suggested that the interaction between tumor and stromal cells may facilitate OC cell spread within the peritoneal cavity [7,8]. There is increasing evidence that adipose tissue, the main tumor microenvironment (TME) component in OC, plays an important role in promoting cancer initiation and progression [9]. OC cells closely interact with adipocytes in the omentum, leading to significant phenotypic alterations in adipocytes known as cancer-associated adipocytes (CAAs). CAAs are especially important in cancer progression because they directly or indirectly facilitate cancer cell growth, angiogenesis, anti-apoptotic effects, and migration [10]. CAAs are multifaceted, constantly evolving, and play several roles in the construction of a tumor-promoting TME [11].

As one of the biggest risk factors for cancer development, aging promotes cancer progression through a variety of mechanisms, including accumulation of cellular damage, increased systemic inflammation, and attenuated adaptive immunity [12]. Adipose tissue displays species-conserved temporal changes with aging, including redistribution from peripheral to central depots, loss of thermogenic capacity, and expansion within the bone marrow [13]. Adipose tissue is one of the earliest organs to respond to aging, as determined by single-cell transcriptomic analyses, making it possible to subsequently drive whole-organism aging [14]. Therefore, adipose tissue may be a central driver of organismal aging and age-associated diseases [13]. However, the extent to which these changes contribute to OC development remains unclear. Unfortunately, previous studies have been largely limited to the effects of aging in healthy donors or have focused on the impact of the TME in young mice [15]. The importance of aged adipose tissue in the aged TME remains unclear.

Therefore, this study focused on the effects of an aged adipocyte-rich TME surrounding OC on cancer development and progression. The objectives of this study were to investigate how alterations in gene regulatory pathways and metabolism differ between aged and young TME, with a focus on adipose tissue, and how the aged TME might contribute to a more aggressive OC phenotype.

## 2. Results 

### 2.1. Aging Promotes Tumor Formation and Growth

To explore the impact of an aged host on tumor development, young (6 weeks) and aged (12 months) female athymic nude rats (RNU) were implanted into the fat tissues near the ovaries with 5 × 10^6^ A2780 cells. The rats underwent survival surgery to remove the tumors on day 26 after cancer cell implantation and the primary tumors and surrounding fat tissues were collected (Figure 1). Figure 2A shows that 13 out of 13 (100%) aged rats formed tumors, while 11 out of 16 (68.75%) young rats formed tumors (*p* = 0.047). Furthermore, the tumor volumes in the aged rats were significantly higher than that in young rats, with a range from 0.8 to 223.0 cm^3^ (median = 17.6 cm^3^) for tumors in the aged rats and a range from 0 to 21.5 cm^3^ (median = 1.1 cm^3^) for tumors in the young rats (*p* = 0.0023, Figure 2B).

### 2.2. Transcriptome Analysis Shows Differential Gene Expression in PTA-A&Y, TSA-A&Y, and PTA&TSA-A

Transcriptomes of gonadal adipose tissues from aged and young pre-tumor adipose (PTA-A&Y), aged and young tumor surrounding adipose (TSA-A&Y), and aged adipose between pre-tumor and tumor surrounding adipose (PTA&TSA-A) were analyzed using RNA sequencing (RNA-seq). A total of 28,914 transcripts (Appendix A) were analyzed using principal component analysis (PCA) for gene expression. The PCA plots (Figure 3A–C) showed that each sample was clearly distinguished on the score map, and the focus between each reset was tight, indicating differences in the transcripts of PTA-A versus PTA-Y, TSA-A versus TSA-A-Y, and PTA-A versus TSA-A. When the threshold was set to |log2 FC| ≥ 1 and adjusted *p* ≤ 0.05, 2323 differentially expressed genes (DEGs) were identified between PTA-A and PTA-Y, 1002 between TSA-A and TSA-Y, and 2612 between PTA-A and TSA-A (Figure 4A–C. The DEGs in the aged group and genes with no significant changes are also presented in the volcano plots in Figure 4A–C. Figure 5A shows the Venn plot of upregulated and downregulated DEGs in PTA-A&Y, TSA-A&Y, and PTA&TSA-A groups. The heatmap in Figure 5B shows 25 upregulated and eight downregulated genes in aged rats for aged vs. young PTA tissues, TSA tissues, and aged PTA vs. aged TSA. Among these genes, seven (21.21%) were adipokine-related genes, including *S100a8*, *S100a9*, *Mmp8*, *Lcn2*, *Pdpn*, *C3*, and *Pnpla3* (Appendix A). 

Next, we determined if there was enrichment of the different groups of significant DEGs in KEGG pathways or for Gene Ontology terms for the three comparison groups when the threshold was set to the absolute value of the Normalized Enrichment Score |NES| ≥ 1 and *p* ≤ 0.05. Twenty-six KEGG pathways, 45 biological processes, 28 molecular functions, and five cellular components were differentially enriched in all three groups (Figure 6, Appendix A). These analyses revealed the enrichment of several pathways/functions, such as cytokine receptors, cytokines, growth factors, viral protein interaction with cytokine and cytokine receptors, *IL-17* signaling pathway, and glycerolipid metabolism in KEGG. Among the GO Biological Process terms, we identified the enrichment of significant DEGs for the cytokine-mediated signaling pathway, fatty acid metabolic process, immune response, T cell activation, immunoglobulin-mediated immune response, and negative regulation of T cell proliferation. These data suggest that cytokines and cytokine-related pathways, lipid metabolism, and immune cells are involved in the development of OC in aged animals.

### 2.3. Regulation of Immune Microenvironment of OC

TIMER was used to estimate immune cell infiltration in all three comparison groups: PTA-A and Y, TSA-A and Y, and PTA-A and TSA-A. Immune cell infiltration varied significantly among the three groups (Appendix A). Higher infiltration levels of neutrophils, CD4 + T cells (non-regulatory), and myeloid dendritic cells were significantly different in aged rats than in young rats in PTA and TSA, as well as TSA-A compared with PTA-A. Activated mast cell infiltration was higher in aged rats than in young rats before tumor generation (PTA-A vs. PTA-Y) and in aged OC xenografts compared with young OC xenografts when the primary tumor formed (TSA-A vs. TSA-Y). Lower infiltration levels of activated mast cells in aged rats were evident after OC generation (TSA-A) than before tumor formation (PTA-A). 

### 2.4. Lipid Metabolic Involvement in Aged Tumor Generation

The metabolome Orthogonal Partial Least Squares Discriminant Analysis of rat adipose tissues during primary tumor formation is presented in Figure 7A. A total of 28 TSA samples, including 13 aged and 15 young, were clearly distinguished by an orthogonal T of 35.1% and T of 5.1% of the total variability, respectively (Figure 7A). T-scores for the repeated samples within each group (young or aged) were closely related and were clearly separated between the groups, indicating the reliability of the metabolome data. In total, 905 metabolites were identified (Appendix A), including 545 positive and 360 negative ions. According to the Relational database of Metabolomic Pathways functional annotation, 70 pathways were enriched, of which 24 (34.29%) had the largest number of metabolites. The top 25 hits of functional pathways are shown in Figure 7B, which included the top 10 relevant pathways, such as G alpha (q) signaling events; synthesis, secretion, and inactivation of Glucagon-like Peptide-1 (GLP-1); incretin synthesis, secretion, and inactivation; free fatty acid receptors; linoleic acid oxylipin metabolism; peptide hormone metabolism; Class A/1 (rhodopsin-like receptors); G protein-coupled receptor (GPCR) downstream signaling; octadecanoid formation from linoleic acid; and Omega-3/omega-6 fatty acid synthesis. With a threshold of VIP ≥ 1, |log2 FC| ≥ 1, and *p* < 0.05, 37 differentially expressed metabolites (DEMs) between young and aged groups were selected, including 17 upregulated lipids and 20 downregulated lipids in aged rats compared to young rats at the stage of primary tumor formation (Figure 7C, Appendix A). The top 10 significant DEMs were CoenzymeQ8, CoenzymeQ9, Taurocholic acid, Triglycerides (TG) (18:1_20:1_24:1), TG (19:0_18:1_20:1), TG (18:1_20:1_22:5), PE (19:0_18:2), TG (24:1_18:2_18:2), TG (18:1_18:1_26:1) and TG (24:0_18:1_20:1). Figure 7D–F shows the top three DEMs, including thirteen glycerolipids (GLs), ten glycerophospholipids (GPs), and nine fatty acids (FAs).

### 2.5. Transcriptome–Metabolome Networks

Next, we explored the correlation between RNA-seq results and lipid metabolites from the tumor surrounding adipose (TSA) tissue when the primary tumor was generated. The gene–lipid correlations in TSA_A&Y were evaluated using Spearman’s correlation. We defined DEGs as those with a correlation of r ≥ 0.5, |log2 FC| ≥ 1 for gene expression, and *p* < 0.05; 207 DEGs and 67 DEMs were identified (Figure 8A). Among these, punicic acid (9E,11Z,13E)-9,11,13-Octadecatrienoic acid (FFA (18:3)) and phosphatidylserine (16:0/22:6 (4Z,7Z,10Z,13Z,16Z,19Z)) was the best explanatory variable in the fatty acid metabolic process and glycerolipid metabolism (correlated to *Pnpla3* gene). LysoPE (0:0/22:5(4Z,7Z,10Z,13Z,16Z)) (LPE (0:0/22:5)), lysophosphatidylethanolamine, and lysophospholipid were the best explanatory variables for the cytokine receptors (correlated to the gene). FFA and TG were the best explanatory variables in the *IL-17* signaling pathway, with correlations to *S100a8* and *S100a9* (r ≥ 0.5 and *p* < 0.05). FFA (18:3) was correlated to genes of *Pnpla3*, *S100a8*, and *C3* (r ≥ 0.5 and *p* < 0.05). A more detailed description of the variables is provided in Appendix A. 

We also explored the correlation between immune infiltration and gene expression from RNA-seq and lipid metabolites from TSA tissues when the primary tumor was generated, as shown in Figure 8B for gene–immune cell infiltration and Figure 8C for lipid–immune cell infiltration. We used a correlation threshold of r ≥ 0.5 and a significance threshold of *p*< 0.05 for gene versus immune infiltration and lipid versus immune infiltration. We found that infiltration levels of neutrophils were related to the expression of *S100a8*, *S100a9*, *C3*, *Fcna* and *Pnpla3*, and lipids 13-oxoODE, Cer (t18:1/18:0(2OH)), CoenzymeQ8, CoenzymeQ9, PE(19:0_18:2), Taurocholic acid, TG (18:1_20:1_22:5), TG (18:1_20:1_24:1), TG (19:0_18:1_20:1), 9,10-DiHOME, and FFA (18:3). The infiltration levels of T cell CD4+ (non-regulatory) were related to *S100a8* and *S100a9*, and lipids CoenzymeQ8, CoenzymeQ9, Taurocholic acid, and TG (18:1_20:1_24:1). The infiltration levels of myeloid dendritic cells were related to genes *S100a8*, *S100a9*, *C3*, *Fcna* and *Pnpla3*, and lipids 9,10-DiHOME, (±)12-HEPE, 13-oxoODE, Cer (t18:1/18:0(2OH)), FFA (18:3), CoenzymeQ8, CoenzymeQ9, PE (19:0_18:2), Taurocholicacid, TG (18:1_20:1_22:5) and PS (16:0_22:6). The infiltration levels of mast cell activation were related to genes *S100a8* and *S100a9*, and lipids CoenzymeQ8, CoenzymeQ9, TG (18:1_20:1_22:5), TG (18:1_20:1_24:1), and TG (19:0_18:1_20:1). Additionally, FFA (18:3) levels were correlated with neutrophils and myeloid dendritic cells. A more detailed description of the variables is presented in Appendix A. 

To validate the in vivo findings, we performed an OC cell growth study using aged conditioned medium (ACM) from aged 3T3-L1 preadipocytes. While aged 3T3-L1 cells showed intensive beta-gal staining (Figure 9A), suggestive of senescence and cell aging, A2780 cells showed faster cell proliferation when exposed to ACM from aged 3T3-L1 cells (Figure 9B). Our data indicated that the aged adipose microenvironment is more vulnerable to OC outgrowth. We performed RT-qPCR analysis (Figure 9C) to validate gene expression. For all eight genes that showed significant involvement in lipid regulation, including *Pnpla3* (glycerolipid metabolism), *Il1rl1* (cytokine receptor), *S100a8*, *S100a9*, and *Lcn2* (IL-17 signaling pathway). Consistent with the RNA-seq data, *S100a8*, *S100a9*, *C3*, *Il1rl1*, *Lon2*, *Hba-a1*, and *Fcna* exhibited increased expression while *Pnpla3* showed decreased expression in adipose tissues from aged rats as compared to the adipose tissues from young rats (Figure 9C). Our lipidomic analysis showed a negative correlation between FFA (18:3) levels and the more aggressive OC phenotype observed in aged animals. Therefore, we assessed the role of FFA (18:3) in A2780, HEYA8, and CAOV2 OC cell lines by adding FFA (18:3) to the cell culture medium at the concentrations indicated in Figure 9D In agreement with the in vivo data, the addition of FFA (18:3) reduced OC cell proliferation (Figure 9D) in all three lines tested. 

## 3. Discussion

This study explored the functional contribution of an aged TME to OC development, with a focus on adipose tissue. Tumor generation increased in aged animals, with higher tumor formation rates and larger tumor volumes. We revealed significant shifts in gene expression enrichment and lipid components of gonadal adipose tissues using RNA-seq and lipidomics analysis. We also found an increased infiltration level of immune cells in the aged versus young TME, including neutrophils, CD4+ T cells (non-regulatory), and myeloid dendritic cells. Correlations were observed between the infiltration of immune cell types and gene expression. Finally, we used gene expression profiling from RNA-seq data to identify 26 chemotherapeutic drugs that may be more effective against OC in older women.

Ovarian cancer occurs in postmenopausal women and is rare in women below the age of 40 years. This study used 12-month-old rats to mimic postmenopausal women because studies have shown that rats typically experience irregular estrous cycles (termed estropause) around the age of 9–12 months due to the disruption of hormones and the resulting reproductive decline [16]. Athymic nude rat xenograft models provide several advantages in aging and metastasis studies compared with mouse models. The rat model has larger abdominal fat pads (a common site of OC metastasis), a more robust tolerance to chemotherapy and surgery, a longer lifespan, the ability to harvest a larger volume of blood/tissue for downstream analyses, and pharmacokinetics that are closer to those of humans [17]. More importantly, the metabolic physiology of rats is more closely aligned with that of humans than with that of mice [18], making research findings more translatable to humans. Athymic nude rats have been widely accepted as model organisms because of their ability to grow many tumor cell lines of human and rodent origin for several months [17,19]. We used 6-week-old female athymic nude rats, which are sexually mature [20,21], as the young study group, and 12-month-old rats, which are post-estropause, as the aged study group.

In humans, ovarian cancers develop in the ovaries and fallopian tubes, regions that contain minimal adipose tissue. However, most ovarian cancers spread rapidly through exfoliation to the omentum, which is rich in adipose tissue and serves as the tumor microenvironment (TME). The TME is a heterogeneous ecosystem composed of infiltrating immune cells, mesenchymal support cells, and matrix components, all of which contribute to tumor progression. Adipose tissue, a primary cellular component of the metastatic OC TME, has not been extensively studied in the progression of solid cancers, particularly OC. We first observed greater tumor formation and larger tumor volumes in aged animals than in young animals. Our finding that aged animals supported the growth of higher numbers of larger tumors was also supported by a cell culture model that showed that senescent preadipocytes (3T3-L1) promoted the proliferation of OC cells compared with young cells. These data suggest that it is primarily adipose tissue in the aged tissue microenvironment that provides favorable “soil” for cancer development and growth. 

Studies have shown that the infiltration of inflammatory cells and enlargement of lipid droplets in adipocytes are more pronounced in aged adipose tissues [22]. Compared with mature adipocytes, aged adipocytes exhibit increased senescence and significant enhancement of pro-inflammatory cytokines and ECM components [23]. Progenitor cells (e.g., adipose tissue stem cells or stromal progenitor cells) in adipose tissues are among the most responsive to aging and aging-related environments [13]. Adipose tissue-derived mesenchymal stem cells from older donors (humans and mice) exhibit senescence and impaired regenerative potential [24]. In addition, although increasing adipocyte size allows for increased lipid storage with aging, larger adipocytes are more insulin-resistant than smaller adipocytes, leading to enhanced rates of basal lipolysis [25]. Inefficient storage of lipids and increased basal lipolysis in hypertrophied adipocytes normally result in exposure to increased levels of fatty acids in non-adipose cells in the vicinity. Excess fatty acids have a negative impact on systemic energy homeostasis. This imbalance may favor tumor growth in older humans. A growing number of studies have also found that aged hosts are more susceptible to cancer metastasis [3], which supports our findings that aged animals exhibit more aggressive cancer. Our study provides strong evidence of the molecular mechanisms underlying these cellular changes. 

Unlike other lipids that appear to promote tumor development, FFA (18:3), which was decreased in aged rats (Figure 7F), inhibited OC cell proliferation (Figure 9D). FFA (18:3), (9E,11Z,13E)-9,11,13-Octadecatrienoic acid, is a punicic acid, also known as 9t,11C,13t-CLN or pumicate, belonging to omega-5 and the class of organic compounds known as lineolic acids and their derivatives. FFA (18:3) has been reported to have anticancer properties in vitro in breast cancer (MCF-7), lung cancer (A549), colorectal cancer (DLD1), stomach cancer (MKN-7), and liver cancer (HepG2) [26]. FFA (18:3) can induce ferroptosis as a single agent via a mechanism distinct from that of canonical ferroptosis inducers [27]. Chou et al. reported that the lower adipogenicity effect of FFA (18:3) in 3T3-L1 cells was mediated by the induction of apoptosis in preadipocytes [28]. The specific cytotoxicity of FFA (18:3) in pre- and post-confluent preadipocytes to induce apoptosis may contribute to the repression of adipocyte differentiation [28]. We found that aged 3T3-L1 cells promoted OC cell proliferation compared to non-senescent 3T3-L1 cells (Figure 9A,B). Whether FFA (18:3) downregulation in the aged TME is due to reduced synthesis in senescent 3T3-L1 cells requires further investigation. Taken together, FFA (18:3) may be a potential chemopreventive and therapeutic agent for ovarian cancer in elderly patients. Further research is required to assess its mechanisms of action and potential clinical applications. 

In addition to exploring lipid metabolism, we examined the transcriptome of TME adipose tissue in young and aged animals with and without OC. Previous single-cell transcriptomics and RNA-seq studies have shown that differential gene expression arises in gonadal adipose tissue mid-life, before in other organs, and persists with advancing age, exhibiting firm aging profiles [14]. This suggests that the change in adipose gene expression is an early event in life and is sensitive to aging. In this study, we identified twenty-five and eight genes with increased and decreased expression, respectively, in all three comparisons. These results support functional contributions to adipose aging and tumor development in aged individuals. Correlation analysis linked DEGs and DEMs, further emphasizing the importance of DEGs in lipid metabolism. Adipocytes secrete over 600 metabolites, hormones, and cytokines, collectively known as adipokines [29]. Our candidate genes are mostly adipokine-related genes, except for *Il1rl1* (interleukin 1 receptor like 1) and *Hba-a1*, whose protein product functions are related to proinflammation and abnormal aerobic respiration, respectively. 

Calprotectin, *S100a8* and *S100a9* are calcium- and zinc-binding proteins belonging to the *S100* family. They play a prominent role in the regulation of inflammatory processes and immune responses in diseases including cancer [30]. They are abundant in the cytosol of neutrophils and can induce chemotaxis and adhesion [30]. *S100a8*, *S100a9*, and the *S100a8*/a9 heterodimer are involved in cancer development, progression, and metastasis by interfering with tumor metabolism and microenvironment [30]. High expression of *S100a8* and *S100a9* has been reported in ovarian cancer [31,32]. Our results showed that *S100a8* and *S100a9* are highly expressed in the TME of aged animals with ovarian cancer and may contribute to a more aggressive cancer phenotype in these animals. A previous study reported that the *S100a8*/a9 heterodimer can bind to polyunsaturated fatty acids, such as oleic, linoleic, and arachidonic acids, in a calcium-dependent manner [33]. Consistent with these data, our results show that the expression of *S100a8/a9* in the TME of aged animals is correlated with TG and various fatty acids, such as FFA (12:0), FFA (15:0), and FFA (18:3). Future work will focus on how cancer cells are influenced by *S100a8*/a9 regulated lipids and the role of *S100a8*/a9 in cancer cell metabolism via the mitochondria. 

A previous study suggested that the release of *S100A8*/*A9* proteins and lipids modified by neutrophils can induce reactivation of dormant tumor cells (cancer stem-like cells) [34], further emphasizing the importance of *S100A8*/*A9* proteins in the TME for cancer development and recurrence. Resolving the molecular mechanisms mediated by *S100A8/A9* may reveal opportunities for designing novel cancer therapeutics. Other adipokine-related genes, including *Lcn2*, *C3*, *Fcna*, and *Pnpla3*, were also identified in our study because of their association with lipids in the aged OC model. These genes are worthy of further study for their role in OC genesis.

Our data suggest that age-related changes in immune cell composition, particularly inflammation-related cells in adipose tissue, may contribute to the efficiency of proliferation in aged animals. Chronic inflammation significantly impacts the overall health and immune function of older cancer patients. Chronic inflammation is also an important characteristic of aging. It can accelerate disease progression in many types of cancer and is often exacerbated by conventional cancer treatments. Obesity and aging lead to alterations in visceral adipose tissue, characterized by both quantitative and qualitative changes in infiltrating and resident immune cells. These changes shift towards a pro-inflammatory phenotype, with immune and stromal progenitor cells, rather than adipocytes, serving as the primary sources of pro-inflammatory mediators that contribute to age-related adipose tissue inflammation [35,36]. Our data revealed that age-related changes in immune cell composition, especially inflammation-related cells in adipose tissues, might enhance cancer cell proliferation in aged animals. We found that gene expression, lipid components, and immune cell composition were interconnected in aged animals with cancer, indicating their functional interdependence. Lipid changes resulting from aging-associated alterations in gene expression (such as *S100a8/a9*) could initiate immune cell infiltration, shifting the local environment towards a pro-inflammatory phenotype [35,36]. This is supported by evidence of an inflammatory pattern present in obese and aging adipose tissue, with abundant accumulation of M1 macrophages, neutrophils, mast cells, B2 cells, CD8+ T cells, and Th1 cells [37]. Neutrophils facilitate the formation of the ovarian cancer pre-metastatic niche in the omentum [38]. Other studies have suggested that neutrophil extracellular trap formation correlates with favorable overall survival in high-grade ovarian cancer [39]. However, a substantial body of evidence from mouse models indicates that tumor-infiltrating neutrophils can exhibit both tumor-promoting and antitumor functions [40,41]. Further studies on tumor-infiltrating neutrophils and their regulation in relation to aging and cancer are required.

The current study has several limitations. First, a major disadvantage of using athymic nude rats as a model system is the lack of functional immune cells, which affects the TME in omental fat. Second, this study used 12-month-old rats as aged animals because, around the age of 9–12 months, rats typically experience irregular estrous cycles due to hormone disruptions and a resulting reproductive decline [16]. Future studies using older animals, such as two-year-old rats, would better mimic human OC patients who are primarily affected in their 60s. Third, as a preliminary step, we report a network of genes, lipids, and immune cells in aged rats with OC relative to controls. Further mechanistic studies using in vivo and in vitro models are required. 

## 4. Materials and Methods

### 4.1. Rat Xenograft OC Model 

The human ovarian adenocarcinoma cell line A2780 was used to generate xenograft tumors in rats, as previously used for tumor generation in 6–8-week-old female athymic nude rats (RNU) purchased from Charles River Laboratories (Wilmington, MA, USA). All animal experimental protocols were approved by the Institutional Animal Care and Use Committee (IACUC, Protocol Registry Number A132-21-06). All methods were reported in accordance with the ARRIVE guidelines. All rats were housed at room temperature, maintained within a range of 21–24 °C, with humidity levels of 50 ± 2% and a 12-h light–dark photoperiod. For the “aged” (post-estropausal) group, six-week-old RNU rats (n = 16) were raised in Duke University’s Division of Laboratory Animal Research (DLAR) animal facility until they reached 12 months of age. During this period, body weight, health conditions, and behavior were assessed weekly. Three rats died over the study period, leaving 13 rats in the aged group. Then, sixteen 6–8-week-old female athymic nude rats (RNU) were purchased from Charles River Laboratories. These aged (12-month-old) and young (6-week-old) rats underwent surgery simultaneously. A2780 cells (5 × 10^6^ per rat) were prepared by embedding them in 100 µL of Growth Factor Reduced Basement Membrane Matrix (Cat#354230, Corning, Durham, NC, USA) and kept on ice to maintain the matrix in a liquid state. The surgery was conducted under sterile conditions at the DLAR facility. Two skin incisions, each approximately 2 cm long, were made on the backs of the rats, beneath the rib cage and about 1 cm away from the spine. The muscular layer was carefully elevated using forceps to avoid damaging the underlying tissues. The peritoneal wall directly beneath the cutaneous incision was incised to expose the ovaries and surrounding adipose tissue. The A2780 cells (5 × 10^6^ per rat) were injected using a 1 mL syringe directly into the adipose tissue surrounding the right ovaries of both aged and young rats. A piece of adipose tissue (approximately 1 cm^3^) surrounding the left ovaries, referred to as “pre-tumor adipose” (PTA), was excised for RNA-seq analysis. The skin incisions were closed with 2 or 3 wound clips, which were removed one week post-surgery. After implantation, the rats were examined daily for tumor formation through visual inspection and abdominal palpation. At 26 days post-cancer cell implantation, all aged and young rats underwent a survival surgery using a procedure similar to that employed for the initial cancer cell implantation, conducted under sterile conditions in the DLAR facility. Briefly, 2 cm long skin incisions were made on the right side of the rats’ backs, beneath the rib cage and approximately 1 cm away from the spine. The muscular layer was carefully elevated using forceps, and the peritoneal wall was incised to open the peritoneal cavity. The tumors and all adipose tissue surrounding the tumors, referred to as “tumor surrounding adipose” (TSA), were excised using surgical forceps and scissors. The skin incision was closed with 2 or 3 wound clips, which were removed one week after the surgery. Tumor length and width were measured using calipers (Cat # 36934-154, VWR, Radnor, PA, USA), and tumor volumes were calculated using the formula V = 0.5 × L × W^2^, where V is the tumor volume, L is the tumor length, and W is the tumor width.

### 4.2. Cell Lines

All ovarian cancer cell lines and mouse preadipocyte 3T3-L1 cell line (Cat # CL-173) were purchased from the American Type Culture Collection (ATCC) (Manassas, VA, USA). OC cells were maintained in RPMI 1640 (Cat # A4192301, Thermo Fisher, Norristown, PA, USA) supplemented with 10% fetal bovine serum (FBS) (Cat# A5256801, Thermo Fisher, Norristown, PA, USA) and 1% penicillin-streptomycin (*p*/S, Cat #p4333, MilliporeSigma, Darmstadt, Germany). 3T3-L1 cells were maintained in DMEM with high glucose (Cat # D5796, MilliporeSigma, Darmstadt, Germany), 10% bovine calf serum (BCS, Cat # 12133C, MilliporeSigma, Darmstadt, Germany), and 1% P/S. Cells were incubated at 37 °C in a humidified chamber with 5% CO_2_. All cell lines were genetically authenticated before use by the Duke University DNA Analysis Facility and were confirmed to be mycoplasma-free by the Duke Cell Culture Facility.

### 4.3. Senescence of 3T3-L1 Cells

The senescence of 3T3-L1 cells was induced by 2 μM carboplatin (Cat# C2538, MilliporeSigma, Darmstadt, Germany) in growth medium for 72 h, as previously described [42,43]. The cells were washed with PBS and maintained in the growth medium for 24 h for recovery. The cells were then passaged and incubated for six days before use. This exposure strategy was selected based on conditions that did not elicit robust cell death (<10–15%), which resulted in metabolic changes associated with cellular senescence.

Senescence in 3T3-L1 cells was assessed using a Senescence-associated β-galactosidase (SA-β-Gal) Staining Kit according to the manufacturer’s instructions (Cat#9860, Cell Signaling Technology, Inc., Danvers, MA, USA). The cells were stained overnight in a cell culture dish at 37 °C in an incubator without CO_2_. Images were captured using an EVOS M7000 microscope (Thermo Fisher Scientific) at a 10 × 10 magnification.

### 4.4. Aged Conditioned Cell Culture and Cell Proliferation Assay 

A2780 cells were cultured in conditioned medium (CM) from young and aged 3T3-L1 cells in 96-well plates at 2000 cells/well. Carboplatin-unexposed 3T3-L1 cells (young CM) were mixed with carboplatin-exposed 3T3-L1 cells (aged CM) in a 1:2 ratio. A2780 cell proliferation was analyzed 48 h after culture in CM using the CellTiter One Solution Cell Proliferation Assay kit according to the manufacturer’s instructions using 3-(4,5-dimethylthiazol-2-yl)-5-(3-carboxymethoxyphenyl)-2-(4-sulfophenyl)-2H-tetrazolium assay (MTS assay, Cat# G8461, Promega, Durham, NC, USA). The luminescence signals were recorded using a plate reader (BMG Labtech, POLARstar Omega, Offenburg, Germany). The readings were normalized to those of the cells cultured in young CM and reported as the percentage of viable cells. This test was independently repeated three times.

### 4.5. Free Fatty Acid (FFA 18:3) Treatment

The human OC cell line A2780 and high-grade serous epithelial OC cell lines HEYA8 and CAOV2 were cultured with FFA 18:3 9(Z),11(E),13(Z)-Octadecatrienoic Acid, an ω-5 long-chain polyunsaturated fatty acid (Cat# AA00DIYY, Cayman, Ann Arbor, MI, USA) at the indicated dose ranges (2000/well for A2780, 1000/well for HEYA8, and 2000/well for CAOV2) in 96-well plates. Cell proliferation was analyzed after 48 h of culture. The readings were normalized to cells without FFA (18:3) treatment and reported as the percentage of viable cells. This test was independently repeated three times.

### 4.6. RT-qPCR 

RNA was isolated from 20 mg of adipose tissue using the TRI Reagent^®^ (Cat# AM9738, Thermo Fisher, Norristown, PA, USA), according to the manufacturer’s instructions. RT-PCR was carried out using 300 ng of total RNA in a 20 µL reaction volume using the SuperScript IV One-Step RT-PCR kit according to the manufacturer’s protocol (Cat# 12594025, Thermo Fisher, Norristown, PA, USA) with Taqman probes (Thermo Fisher, Norristown, PA, USA) specific to rat genes (Appendix A). GAPDH served as an endogenous control for the RNA input. PCR was performed at 50 °C for 10 min for DNA synthesis, followed by 95 °C for 1 min and 45 cycles at 95 °C for 10 s and 60 °C for 1 min. Relative RNA expression values were calculated using delta-delta CT values and normalized to GAPDH CT values. This test was independently repeated three times with replicates for each condition in each test.

### 4.7. RNA Sequencing

RNA extraction was performed using TRI Reagent^®^ (Thermo Fisher, Norristown, PA, USA), and RNA sequencing (RNA-seq) was performed by Admera Health (South Plainfield, NJ, USA) with RNA from 20 mg of fat tissues, including 29 PTA (16 young and 13 aged rats) and 28 TSA (16 young and 12 aged rats) tissues collected as described above in the section of the Rat Xenograft Model. The full protocol is described in the Appendix A.

### 4.8. Quantitative Lipidomics Assay

Quantitative lipidomics analysis, which provides absolute quantification of up to 3000 lipid compounds, was performed by Metware Biotechnology Inc. (Woburn, MA, USA) with TSA tissues, including 13 adipose tissues from aged rats and 18 adipose tissues from young rats, using ~20 mg of adipose tissue per rat. The details are described in the Appendix A.

### 4.9. Statistical Analysis

Cell proliferation was evaluated using Student’s *t*-test with GraphPad Prism 10 software (GraphPad Software, LLC., CA, USA). RT-qPCR for gene expression was analyzed and compared between groups using the Mann–Whitney test. The tumor formation rate in rats was compared between the aged and young rat groups using *Fisher’s* exact test, and tumor volumes were analyzed using the *Mann–Whitney* test, *p* < 0.05.

No new algorithms were developed for this study. All statistical analyses of the transcriptome and metabolome data were performed using the Bioinforcloud platform (http://www.bioinforcloud.com/BioinforCloud, accessed on 1 April 2024), TIMER Version 2.0 (http://timer.comp-genomics.org/), and MetaboAnalyst 6.0 (https://www.metaboanalyst.ca/), which were applied by calling the appropriate R package (R-4.4.2). The details are presented in Appendix A.

## 5. Conclusions

In conclusion, this study is the first to characterize the aging adipocyte-rich microenvironment landscape that favors OC, including its transcriptional patterns, tumor immunity profiles, and lipid metabolism features. We demonstrated that aged hosts were more susceptible to OC proliferation in both in vitro and in vivo models. We identified a functional network of OC gene–lipid metabolism-immune cells in aged animals. *S100a8* and *S100a9*, secreted by adipocytes, preadipocytes, or neutrophils, have been identified as key genes affecting lipid metabolism, such as FFA (18:3), and in turn, cancer development in aged animals. This study is important because it provides fundamental information about the genetic and metabolic landscape in the TME, which changes with age and may contribute to OC vulnerability in the aging global population. Evidence obtained from this study supports the contention that interventions to restore a normal functioning microenvironment may reverse age-related adipose tissue dysfunction and enhance treatment strategies in older patients with ovarian cancer.

## Figures and Tables

**Figure 1 ijms-26-00320-f001:**
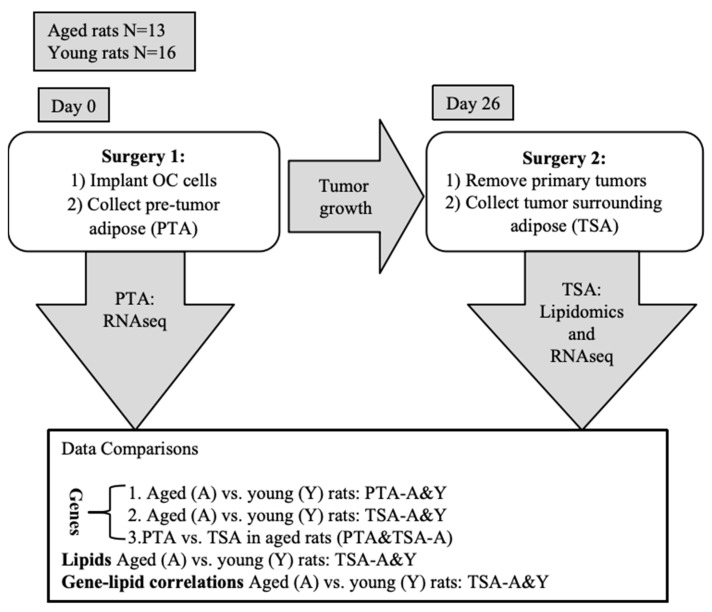
Workflow for xenograft rat model. OC: ovarian cancer. PTA-A: pre-tumor adipose from aged rats. PTA-Y: pre-tumor adipose from young rats. TSA-A: tumor surrounding adipose from aged rats. TSA-Y: tumor surrounding adipose from young rats.

**Figure 2 ijms-26-00320-f002:**
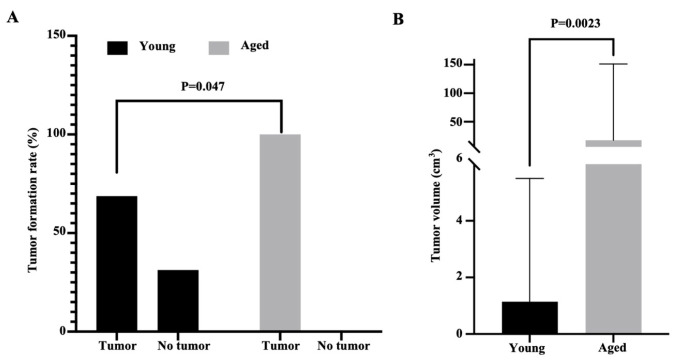
More tumor formation in aged rat group. (**A**). Comparison of the tumor formation rate between aged and young rat groups. Thirteen out of thirteen rats in the aged group formed tumors by day 26 post-ovarian cancer (OC) cell implantation. Eleven out of sixteen rats (68.75%) in the young group formed tumors by day 26 post-OC cell implantation. (**B**). By day 26 post-OC cell implantation, the average tumor volume from the aged rat group was significantly bigger than that from the young rat group, aged tumor median = 17.64 cm^3^, young tumor median = 1.14 cm^3^.

**Figure 3 ijms-26-00320-f003:**
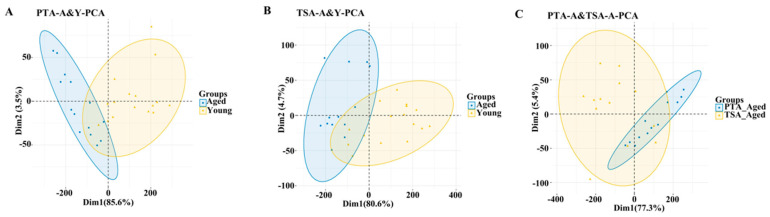
**Principal component analysis (PCA) in PTA-A&Y, TSA-A&Y, and PTA&TSA-A.** PCA was performed with the 29 biological samples including 16 fat tissues from young rats and 13 from aged rats. The analysis showed the similarities of differential gene expressions within the aged groups and the clear separations of gene expression between aged and young groups for all 3 panels of datasets, PTA-A&Y (**A**), TSA-A&Y (**B**), and PTA&TSA-A (**C**). PTA-A: pre-tumor adipose from aged rats. PTA-Y: pre-tumor adipose from young rats. TSA-A: tumor surrounding adipose from aged rats. TSA-Y: tumor surrounding adipose from young rats.

**Figure 4 ijms-26-00320-f004:**
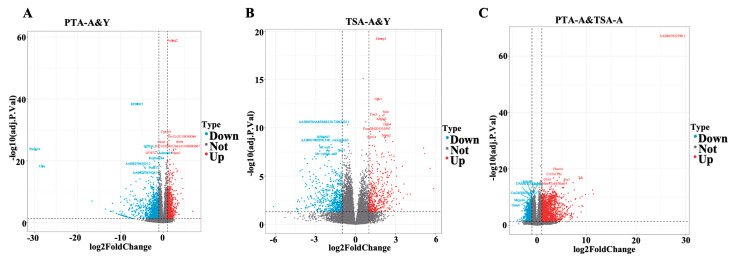
Volcano plots of transcriptome analysis show differential gene expression in PTA-A&Y (**A**), TSA-A&Y (**B**), and PTA&TSA-A (**C**). Comparisons by Volcano plots of unique differential expressed genes (DEGs) in PTA-A&Y, TSA-A&Y, and PTA&TSA-A groups. The x and y axes show the log2 FC and -log10(adjusted *p*-value), respectively. Types down, not, and up indicate downregulated DEGs, no difference DEGs, and upregulated DEGs in the aged group. PTA-A: pre-tumor adipose from aged rats. PTA-Y: pre-tumor adipose from young rats. TSA-A: tumor surrounding adipose from aged rats. TSA-Y: tumor surrounding adipose from young rats.

**Figure 5 ijms-26-00320-f005:**
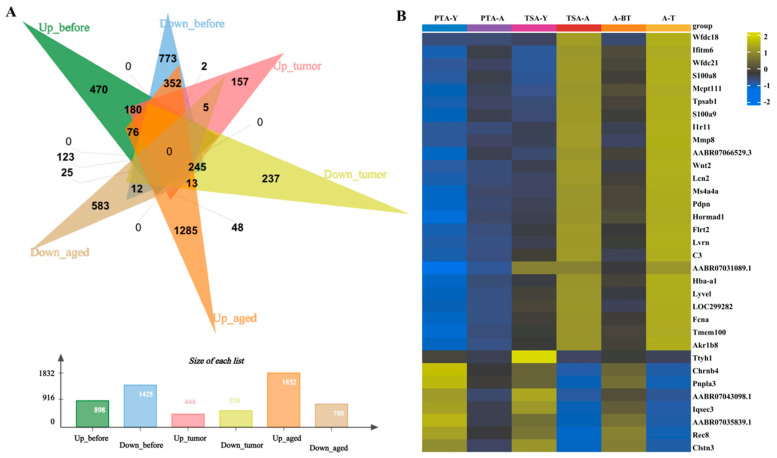
**Transcriptome analysis shows differential gene expression in PTA-A&Y, TSA-A&Y, and PTA&TSA-A.** (**A**). Venn plot of upregulated and downregulated differential expressed genes (DEGs) in PTA-A&Y, TSA-A&Y, and PTA&TSA-A groups. Up-before: upregulated genes in PTA-A&Y; down-before: downregulated genes in PTA-A&Y; up-tumor: upregulated genes in TSA-A&Y; down-tumor: downregulated genes in TSA-A&Y; up-aged: upregulated genes in PTA&TSA-A; down-aged: downregulated genes in PTA&TSA-A. The bar graph shows the number of DEGs in each adipose group. The numbers are included in each group. (**B**). The heat map of DEGs in PTA-A&Y, TSA-A&Y, and PTA&TSA-A. The differentially expressed genes are shown on the right of the heat map. PTA-Y: pre-tumor adipose from young rats. PTA-A: pre-tumor adipose from aged rats. TSA-Y: tumor surrounding adipose from young rats. TSA-A: tumor surrounding adipose from aged rats. A-BT: aged rats before tumor formation. A-T: aged rats when tumor formed.

**Figure 6 ijms-26-00320-f006:**
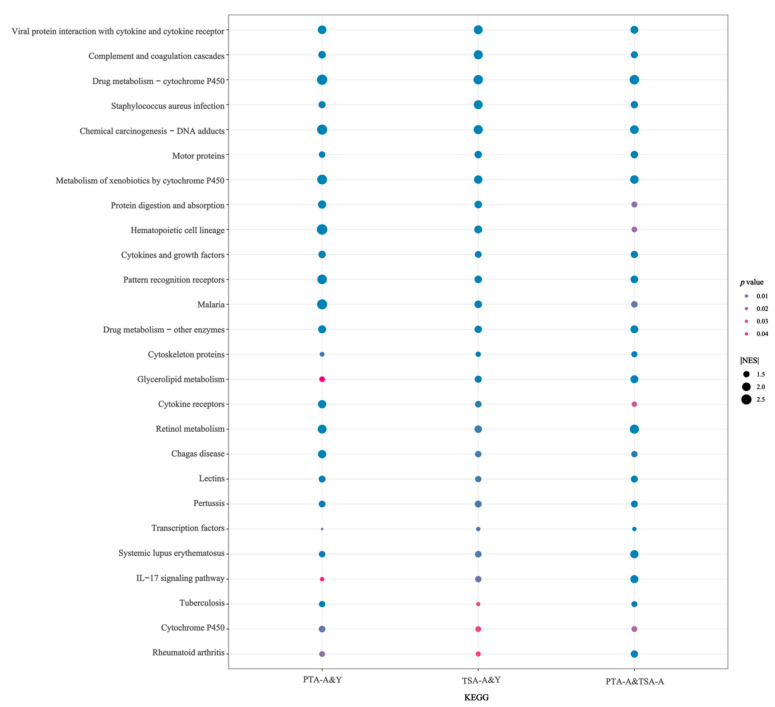
**Enrichment of KEGG Pathway in PTA-A&Y, TSA-A&Y, and PTA&TSA-A groups.** Twenty-six KEGG pathways were enriched in all 3 of the PTA-A&Y, TSA-A&Y, and PTA&TSA-A groups, including cytokine receptors, cytokines and growth factors, viral protein interaction with cytokine and cytokine receptor, IL-17 signaling pathway, glycerolipid metabolism. PTA-A&Y: comparison of pre-tumor adipose from aged rats and young rats. TSA-A&Y: comparison of tumor surrounding adipose from aged rats and young rats. PTA-A: pre-tumor adipose from aged rats. TSA-A: tumor surrounding adipose from aged rats.

**Figure 7 ijms-26-00320-f007:**
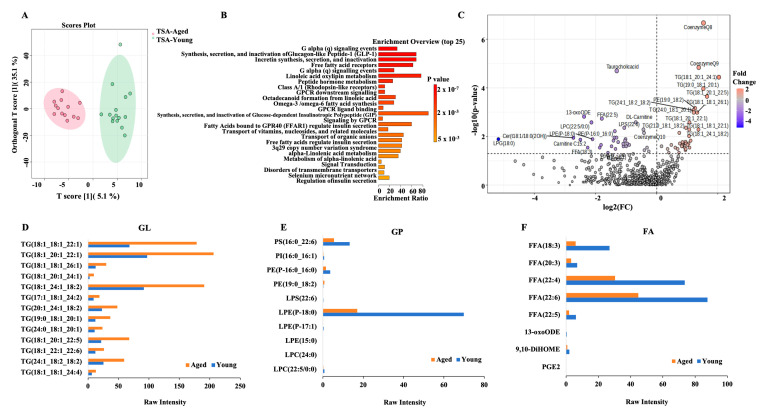
**Differential expression of metabolites in TSA-A&Y adipose.** (**A**). Orthogonal Partial Least Square Discriminant Analysis (OPLS-DA) was carried out with 28 biological samples including 13 adipose tissues from aged rats and 15 adipose tissues from young rats at primary tumor formation (5 young rats without tumors). The green plot shows the lipid components in young rats and the red shows the lipids in aged rats. The lipid samples within each group were closely related, while the separation of lipids between the groups was far, indicating the reliability of the metabolome data from the aged rat adipose groups. (**B**). The top 25 pathways of Relational Database of Metabolomics Pathways (RaMP) functional annotation. (**C**). Volcano plots showed differential expression of metabolites (DEMs) between TSA-A and TSA-Y. The x axes show the log2 FC and the y axes show -log10 (*p*-value), respectively. The dots represent the lipid components in each group. (**D**–**F**). The bar graphs show scaled intensity values of altered glycerolipids (GLs, **D**), glycerophospholipids (GPs, **E**) and fatty acyls (FAs, **F**) between TSA-A (orange bar, aged) and TSA-Y (blue bars, young).

**Figure 8 ijms-26-00320-f008:**
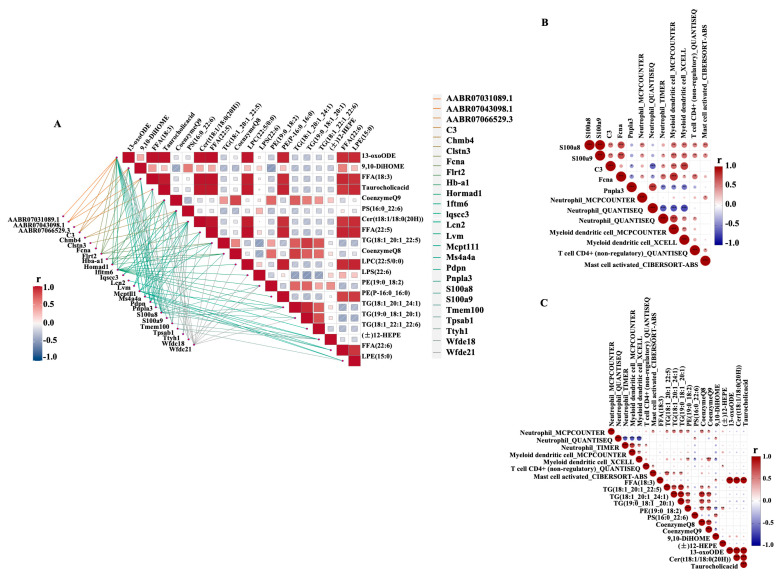
**Transcriptome–metabolome correlative regulation.** (**A**). The gene expression profiling from RNA-seq analysis and lipid profiling from lipidomic analysis were conducted for their correlation in groups of tumor-surrounding adipose from aged rats and young rats (TSA-A&Y), the adipose from primary tumor surrounding tissues from aged and young rats. The gene–lipid, immune cell–gene, and immune cell–lipid correlations were evaluated in TSA-A&Y using Spearman’s correlation. With the thresholds of correlation ≥0.5, absolute log2 FC ≥ 1, VIP ≥ 1, *p* < 0.05, we found 25 genes and 20 lipids that are closely related. The darker the color, the higher the correlation coefficient between gene expression and lipids. (**B**,**C**). The correlation between immune infiltration and gene expression from RNA-seq and lipid metabolites from Lipidomic in groups of TSA-A&Y. (**B**). The correlation between 4 immune cells (Neutrophil, Myeloid dendritic cell, T cell CD4+ (non-regulatory), and Mast cell activated) and 5 genes (*S100a8*, *S100a9*, *C3*, *Fcna*, *Pnpla3*). (**C**). The correlation between 4 immune cells and 13 lipids.

**Figure 9 ijms-26-00320-f009:**
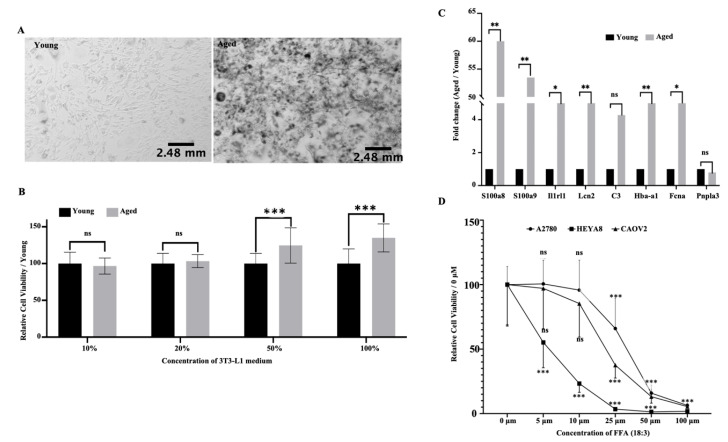
**The in vitro studies confirmed the in vivo findings from rat xenograft model**. (**A**). Aging was induced in preadipocytes, 3T3-L1, with carboplatin treatment at 2 μM for 72 h. The blue color represents senescent cells. (**B**). Ovarian cancer (OC) cell line, A2780, showed higher cell proliferation when exposed in an aging conditioned medium from aged preadipocyte 3T3-L1 cells. ^ns^ *p* > 0.05, *** *p* < 0.001. (**C**). Gene expression was validated using RT-PCR. Eight genes showed differential expression in groups before tumor formation (PTA-A&Y), when tumor formed (TSA-A&Y), and PTA&TSA-A, and their involvement in lipid regulation was validated using RT-PCR with the gene-specific probes for Taqman assay and 6 adipose tissues from each of the aged rats and young rats. As shown with RNA-seq analysis, the genes of *S100a8*, *S100a9*, *C3*, *Il1rl1*, *Lon2*, *Hba-a1* and *Fcna* showed upregulation in aged rats and gene *Pnpla3* showed downregulation in aged rats. ^ns^ *p* > 0.05, * *p* < 0.05, ** *p* < 0.01. (**D**). OC cell line, A2780, HEYA8, and CAOV2 showed lower proliferation when exposed in FFA (18:3) containing medium in a dose-dependent manner. ^ns^ *p* > 0.05, *** *p* < 0.001. OC: ovarian cancer.

## Data Availability

RNA-seq data are shown in Appendix A. The metabolomic data are presented in Appendix A. All other raw data generated in this study are available upon request from the corresponding author.

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
