# Peer review of "Regulation of Age-Related Lipid Metabolism in Ovarian Cancer"

_ijms, 2025, doi:10.3390/ijms26010320_

Round 1
Reviewer 1 Report
Comments and Suggestions for Authors
The manuscript entitled "Regulation of Age-Related Lipid Metabolism in Ovarian Cancer" by Feng et al. presents an investigation into the transcriptomic and lipidomic changes occurring within adipose tissues of young and aged nude rats, both under normal and tumor-associated conditions. The study discovers that the tumor uptake rate of an ovarian cancer (OC) cell line (A2780) is significantly higher in the adipose tissue of aged animals compared to younger ones. Through detailed analysis, a range of differentially expressed genes (DEGs) and metabolites (DEMs) were identified, providing valuable insights into the role of adipose tissues at varying ages in the development and progression of OC. However, for the manuscript to meet the standards required for publication, significant revisions are necessary.
Below are suggestions for manuscript revision:
- Figure Quality: Many images are of low resolution, making labels unreadable. Please enhance the image quality, particularly for transcriptomics and lipidomic analysis plots.
- Methods Description: The experimental details need more clarity, especially concerning the rat xenograft study. Please specify the ages of young and aged animals when the OC cells were inoculated. Detail the procedure for tumor cell inoculation, including whether it involves surgically exposing the abdominal cavity and the exact injection site. Is it in the omental fat tissue near the left or right ovary? Additionally, it is mentioned in line 100 that 'aged and young rats underwent survival surgery to remove tumors and collect fat tissues.' Were the animals monitored for metastasis-related survival afterward? If so, include the survival data.
- Bar Plots: Please add individual data points for the animals in the bar plots throughout the paper.
- Figure 1D & 1E: These should be presented as a separate figure. Consider using parental 3T3 cells as controls to demonstrate that the effects of conditioned media from senescent cells are specific to the adipose cell subtype. It could be better to expand the study with the other two OC cell lines tested in Figure 5E. These in vitro analysis (including 5E) can be combined into one figure to be shown at the end of manuscript as validation of the findings from in vivo samples.
- Figure 2A: Presenting a single PCA plot for all four conditions could offer clearer insights. Enhance the volcano plots' quality and label key DEGs. Move Figure 5D (qPCR validation) to Figure 2. Note a typo in line 337: it should be 'RT-qPCR analyses,' not 'RNASeq analysis,' to validate the RNASeq data.
- Figure 3: Consider using dot plots to display enrichment scores and statistical values for indicated pathways and comparisons.
- TIMER Analysis (Section 3.3): This bioinformatics analysis predicts the percentage of infiltrated immune cells in the tissue among four different conditions. A dedicated figure should be made to present these findings.
- Figure 4: Please define the TSA-Y tissues collected from the animals absent of tumors. Considering Figure 1B shows five young rats without OC tumors in the fat tissue, these samples should be specified or subgrouped. For panels 4D, E, F, include data for individual animals.
- Figure 5: The correlation analysis method needs clearer explanation, and it's currently unclear what conclusions readers should draw from these analyses.
- Result 3.6 (Table 1): The use of the OncoPredict package typically analyzes RNASeq data from tumor samples. Its application to tumor-associated fat tissue might be misleading. Consider removing this section to prevent unsupported claims.
Author Response
Reviewer#1
The manuscript entitled "Regulation of Age-Related Lipid Metabolism in Ovarian Cancer" by Feng et al. presents an investigation into the transcriptomic and lipidomic changes occurring within adipose tissues of young and aged nude rats, both under normal and tumor-associated conditions. The study discovers that the tumor uptake rate of an ovarian cancer (OC) cell line (A2780) is significantly higher in the adipose tissue of aged animals compared to younger ones. Through detailed analysis, a range of differentially expressed genes (DEGs) and metabolites (DEMs) were identified, providing valuable insights into the role of adipose tissues at varying ages in the development and progression of OC. However, for the manuscript to meet the standards required for publication, significant revisions are necessary.
1.Figure Quality: Many images are of low resolution, making labels unreadable. Please enhance the image quality, particularly for transcriptomics and lipidomic analysis plots.
Response: We have revised and improved the resolution of the figures.
2.Methods Description: The experimental details need more clarity, especially concerning the rat xenograft study. Please specify the ages of young and aged animals when the OC cells were inoculated. Detail the procedure for tumor cell inoculation, including whether it involves surgically exposing the abdominal cavity and the exact injection site. Is it in the omental fat tissue near the left or right ovary? Additionally, it is mentioned in line 100 that 'aged and young rats underwent survival surgery to remove tumors and collect fat tissues.' Were the animals monitored for metastasis-related survival afterward? If so, include the survival data.
Response: We have included a detailed surgical procedure, as highlighted in the manuscript under the section titled “Xenograft OC Model”.
Yes, we attempted to monitor metastasis-related survival and tumor recurrence following the surgical removal of the primary tumors. However, several rats died after the surgery, which made the statistical analysis unapplicable.
- Bar Plots: Please add individual data points for the animals in the bar plots throughout the paper.
Response: It is challenging to add the individual data points for some bar plots, such as in Figure 2, which illustrate the tumor formation rates for both young and aged rat groups. Since there were no individual data points available, we opted not to include them in order to maintain a consistent format for the bar graphs.
- Figure 1D & 1E: These should be presented as a separate figure. Consider using parental 3T3 cells as controls to demonstrate that the effects of conditioned media from senescent cells are specific to the adipose cell subtype. It could be better to expand the study with the other two OC cell lines tested in Figure 5E. These in vitro analysis (including 5E) can be combined into one figure to be shown at the end of manuscript as validation of the findings from in vivo samples.
Response: We have changed the original Figure 1B and 1C, which present the animal data, to Figures 2A and 2B. We have presented the in vitro analysis as a separate figure (Figure 9), which was included at the end of the manuscript to validate the findings from the in vivo studies. The RT-PCR data for gene expression validation was relocated from Figure 5D to Figure 9C. The Figure 5E was moved to Figure 9D. Thus, the Figure 9 was all for the in vitro validation analysis. This indeed made the figures and results more organized.
- Figure 2A: Presenting a single PCA plot for all four conditions could offer clearer insights. Enhance the volcano plots' quality and label key DEGs. Move Figure 5D (qPCR validation) to Figure 2. Note a typo in line 337: it should be 'RT-qPCR analyses,' not 'RNASeq analysis,' to validate the RNASeq data.
Response: According to the reviewer's suggestion, we have presented a single PCA plot in Figure 3, enhanced the volcano plots in Figure 4, and relocated Figure 5D (qPCR validation) to Figure 9C.
We apologize for the typographical error regarding 'RNASeq analysis.' It has been corrected to 'RT-qPCR analyses.'
Figure 3
Figure 4
- Figure 3: Consider using dot plots to display enrichment scores and statistical values for indicated pathways and comparisons.
Response: According to the reviewer's suggestion, we have utilized dot plots in Figure 3 to illustrate enrichment scores and statistical values for the comparisons.
- TIMER Analysis (Section 3.3): This bioinformatics analysis predicts the percentage of infiltrated immune cells in the tissue among four different conditions. A dedicated figure should be made to present these findings.
Response: Considering the limitations of the layout, we have presented the results of immune infiltration in detail in Supplement Table S5.
- Figure 4: Please define the TSA-Y tissues collected from the animals absent of tumors. Considering Figure 1B shows five young rats without OC tumors in the fat tissue, these samples should be specified or subgrouped. For panels 4D, E, F, include data for individual animals.
Response: We have defined the TSA-Y tissues collected from the animals absent of tumors as “5 young rats without tumors”.
- Figure 5: The correlation analysis method needs clearer explanation, and it's currently unclear what conclusions readers should draw from these analyses.
Response: We have described the correlation analysis method in Supplement S3 and incorporated it into the results and figure legend. The correlations between gene versus lipid levels, immune cells versus gene expression, and immune cell infiltration versus the lipid levels in TSA_A&Y were conducted using Spearman’s correlation.
- Result 3.6 (Table 1): The use of the OncoPredict package typically analyzes RNASeq data from tumor samples. Its application to tumor-associated fat tissue might be misleading. Consider removing this section to prevent unsupported claims.
Response: We agree and have removed this section from the methods, results, and discussion.

Reviewer 2 Report
Comments and Suggestions for Authors
This manuscript centers around the modulation of age - related lipid metabolism within ovarian cancer, delving deeply into the influence of the aging microenvironment on ovarian cancer, with particular emphasis on lipid metabolism and the distribution of immune cells. It offers novel viewpoints and profound insights in this realm, bearing substantial significance for unraveling the pathogenesis of ovarian cancer and devising treatment strategies tailored to elderly patients. However, some aspects of the manuscript need to be improved.
1. At line 50 on page 2, it would be advisable to supplement the exploration of whether there exist other potential factors, apart from treatment disparities, such as age - related physiological alterations, that might impact the development and treatment response of ovarian cancer in elderly patients.
2. In the "2.1. Rat Xenograft OC Model" section, it is appropriate to append a detailed description of the environmental conditions (including temperature, humidity, photoperiod, etc.) during the rearing period of the experimental animals.
3. In the Discussion section, at line 491 on page 13, while the variations in the infiltration levels of multiple immune cells are presented in the Results, the comprehensive analysis in the Discussion regarding how these immune cell changes coordinately influence the development of ovarian cancer is insufficiently profound. The integration between the results and the overall discussion of the pathogenesis of ovarian cancer is not well - established. It is recommended to enhance the logical coherence between the Results and the Discussion, thereby rendering the article's exposition more comprehensive and persuasive.
Overall, with significant revisions and improvements in the areas mentioned above, the manuscript has the potential to make a valuable contribution to the field of ovarian cancer research.
Author Response
Reviewer #2
This manuscript centers around the modulation of age - related lipid metabolism within ovarian cancer, delving deeply into the influence of the aging microenvironment on ovarian cancer, with particular emphasis on lipid metabolism and the distribution of immune cells. It offers novel viewpoints and profound insights in this realm, bearing substantial significance for unraveling the pathogenesis of ovarian cancer and devising treatment strategies tailored to elderly patients. However, some aspects of the manuscript need to be improved.
1. At line 50 on page 2, it would be advisable to supplement the exploration of whether there exist other potential factors, apart from treatment disparities, such as age - related physiological alterations, that might impact the development and treatment response of ovarian cancer in elderly patients.
Response: We have revised this sentence in accordance with the reviewer's suggestion like this: Although treatment disparities in the elderly may contribute to this difference, older individuals may also be more susceptible to disease progression due to metabolic and immune changes associated with aging.
In the "2.1. Rat Xenograft OC Model" section, it is appropriate to append a detailed description of the environmental conditions (including temperature, humidity, photoperiod, etc.) during the rearing period of the experimental animals.
Response: The housing conditions have been incorporated into the section of Rat Xenograft OC Model.
- In the Discussion section, at line 491 on page 13, while the variations in the infiltration levels of multiple immune cells are presented in the Results, the comprehensive analysis in the Discussion regarding how these immune cell changes coordinately influence the development of ovarian cancer is insufficiently profound. The integration between the results and the overall discussion of the pathogenesis of ovarian cancer is not well - established. It is recommended to enhance the logical coherence between the Results and the Discussion, thereby rendering the article's exposition more comprehensive and persuasive.
Response: We have rewritten this section in accordance with the reviewer's suggestions and highlighted the changes in blue within the discussion section.
Additional requirement.
According to the requirements of our Journal, the final article should
include all of the following sections:
- Introduction
- Results
- Discussion
- Materials and Methods
- Conclusions
Response: We have adjusted the structure of the paper to align with the specified order.
We have revised the manuscript and figures in accordance with the reviewers' suggestions. All changes made will not affect the content or framework of the paper. Additionally, some errors were corrected and highlighted in blue, even though they were not mentioned by the reviewers.
Thank you very much,
Zhiqing Huang

Round 2
Reviewer 2 Report
Comments and Suggestions for Authors
The authors have revised the manuscript in accordance with the reviewers' comments, resulting in an overall clearer logic and more distinct data presentation. The author has conducted a meticulous and in-depth analysis of how age-related lipid metabolism and immune cell distribution influence the progression of ovarian cancer, exploring the impact of the ageing microenvironment on ovarian cancer. The author has enhanced the comprehensive analysis of the relationship between changes in immune cell infiltration levels and ovarian cancer development, resulting in a significant improvement in logical coherence. This has made the overall discussion of the article more comprehensive and persuasive. The manuscript has made a substantial contribution to the research on age-related lipid metabolism regulation in ovarian cancer, providing valuable insights into unraveling the pathogenesis of ovarian cancer and developing therapeutic strategies tailored for elderly patients.
Author Response
ijms-3342077
Type of manuscript: Article
Title: Regulation of Age-Related Lipid Metabolism in Ovarian Cancer
Authors: Jihua Feng, Clay Douglas Rouse, Lila Taylor, Santiago Garcia, Ethan Nguyen, Isabella Coogan, Olivia Byrd, Andrew Berchuck, Susan K. Murphy, Zhiqing Huang *
This research paper includes highly well-conducted study results. Additionally, two reviewers provided extensive feedback, and the authors have appropriately addressed these comments. However, before the paper can be accepted, further revisions are required on the following points.
[Major concerns]
1. Human gene nomenclature: Human genes should be written in italics. Examples: Lines 126 and 127, etc. Examples: S100a8 and S100a9 at Line 38; etc.
Response: Due to our inability to locate the line numbers in the revised version, we have added page numbers for this revision. We have attempted to identify the human genes and have italicized them. These changes are highlighted in blue.
2. IRB statement: The date and approval number of the IACUC authorization from the affiliated institution must be included in the manuscript.
Response: This project did not involve human samples; therefore, Institutional Review Board (IRB) approval was not required. We conducted the animal research under the approved Institutional Animal Care and Use Committee (IACUC) protocol. The details of the IACUC protocol are provided as “IACUC, Protocol Registry Number A132-21-06,” which is highlighted in blue on page 390.
Abbreviations: The use of abbreviations when writing a paper has many advantages besides simplicity of expression. To use an abbreviation, first write the abbreviation in parentheses after the full name and then use the abbreviation from Introduction to the final Conclusion. Abbreviations should only be used if they are repeatedly used and if they are not used again, only the full name should be used. In particular, because of the characteristics of IJMS, where Materials and Methods is arranged at the end of the paper, the original words and abbreviations are written in the order they are used from the introduction, and only when the abbreviation is used repeatedly, the abbreviation can be used until the conclusion.
Response: The corrections have been made. To avoid the lengthy title in the results section (lines 88-89), we instead provided the full name along with its abbreviation in lines 90-92. All correlations have been highlighted in blue.
In cases where abbreviations are used within figures or tables, please list these abbreviations along with their corresponding full names in the figure legends or at the bottom of corresponding tables. If there are two or more abbreviations, arrange them in alphabetical order. In this case, non-proper nouns should not have their first letters capitalized.
Response: The corrections have been made and highlighted in blue.
Materials and Methods section - When naming a particular chemical company, you must provide location information such as company name, city and/or state (abbreviation in the USA and Canada) and country. Once you have named a company with the information, you should only mention a company’s name thereafter.
Response: The corrections have been made and highlighted in blue.
English: In the main text of the paper, there are many instances where the first letter of words in abbreviations is capitalized even though they are not proper nouns. Find all such instances and change them to lowercase. Examples: Cytokine at Line 161; Glycerolipid at Line 162; Interleukin 1 Receptor Like 1 at Line 370; Bovine Calf Serum at Line 477; etc.
Response: The corrections have been made and highlighted in blue.
[Minor concerns]
1. Line 439: As far as I know, it is standard practice to use Celsius, not Fahrenheit, when indicating temperatures in scientific papers.
Response: These have been corrected according to the comments and highlighted in blue.
Line 474: Define ATCC.
Response: This has been done.
Line 475: Define FBS.
Response: This has been done.
Figure 9: As far as I know, statistically, *p<0.05, **<0.01, and ***p<0.001 are enough for biological studies. **** p < 0.0001 is not needed. Please seek advice from a biostatistician around you.
Response: We have removed **** p < 0.0001 from Figure 9.
References: Adjust the citation style of the references to conform to the IJMS guidelines, and ensure that any missing page numbers are accurately included. Examples: 39, etc.
Response: All the page numbers have been added.
Overall, the manuscript can be considered to publication after minor revision as indicated above.
Thank you so much.
Sincerely,
Zhiqing Huang MD., PhD., MS
Assistant Professor
Division of Reproductive Sciences
Department of OB/Gyn
Duke School of Medicine
Chesterfield suite 510
701 West Main Str.
Durham, NC27701
(919)-684-1396 (office)
(919)-593-5536 (cell)
Zhiqing.huang@duke.edu
